# Human Regulatory Macrophages Derived from THP-1 Cells Using Arginylglycylaspartic Acid and Vitamin D3

**DOI:** 10.3390/biomedicines11061740

**Published:** 2023-06-17

**Authors:** Hoang Lan Pham, Thi Xoan Hoang, Jae Young Kim

**Affiliations:** Department of Life Science, Gachon University, Seongnam 13120, Gyeonggi-Do, Republic of Korea; phamlan@gachon.ac.kr (H.L.P.); xoanht89-1@gachon.ac.kr (T.X.H.)

**Keywords:** regulatory macrophage, arginylglycylaspartic acid, vitamin D3, anti-inflammation, xenotransplantation

## Abstract

Regulatory macrophages (Mregs) are unique in that they have anti-inflammatory and immunosuppressive properties. Thus, treating inflammatory diseases using Mregs is an area of active research. Human Mregs are usually generated by culturing peripheral blood monocytes stimulated using a macrophage colony-stimulating factor with interferon (IFN)-γ. Herein, we generated Mregs with an elongated cell morphology from THP-1 cells that were stimulated with phorbol 12-myristate 13-acetate and cultured with both arginylglycylaspartic acid and vitamin D3. These Mregs regulated macrophage function, and respectively downregulated and upregulated the expression of pro-inflammatory and immunosuppressive mediators. They also expressed Mregs-specific markers, such as dehydrogenase/reductase 9, even when exposed to such inflammatory stimulants as IFN-γ, lipopolysaccharide, purified xenogeneic antigen, and xenogeneic cells. The Mregs also exerted anti-inflammatory and anticoagulatory actions in response to xenogeneic cells, as well as exerting immunosuppressive effects on mitogen-induced Jurkat T-cell proliferation. Our method of generating functional Mregs in vitro without cytokines is simple and cost-effective.

## 1. Introduction

Macrophages are phagocytic immune cells that play diverse roles in host defense against infectious agents and tissue injury [1]. Macrophages become activated and convert from one type to another according to various environmental stimuli [2]. The two representative types of macrophages are either classically (M1) or alternatively (M2) activated [3]. M1 macrophages can be differentiated from resting macrophages in response to pathogen- (PAMPs) or damage- (DAMPs) associated molecular patterns, which are derived from either infectious agents or injured tissues, respectively [4,5]. They produce pro-inflammatory cytokines, nitric oxide, and reactive oxygen species (ROS) that kill pathogens. M2 macrophages can be induced using a colony-stimulating factor (M-CSF), or interleukins (IL)-4, -10, or -13, and have anti-inflammatory and regulatory properties [6]. M2 macrophages can be further classified into M2a, M2b, M2c, and M2d subtypes, according to their inducing factors and phenotypic characteristics [5]. Interleukins-4 and -13 induce M2a macrophages [7], which secrete IL-10, TGF-β, and arginase-1 [8]. Immune complexes and toll-like receptor (TLR) ligands, such as lipopolysaccharide (LPS), can induce M2b cells [9] which secrete high levels of both IL-10 and IL-1 receptor antagonist (IL-1Ra), as well as low levels of IL-12 [10,11]. Glucocorticoids, IL-10, and TGF-β each induce M2c cells, which can secrete high levels of IL-10 and TGF-β [12]. Interleukin-6 and A2 adenosine receptor agonists induce M2d macrophages, which can secrete IL-10, transforming the growth factor-beta (TGF-β), C-C Motif Chemokine Ligand 22 (CCL22), and vascular endothelial growth factor (VEGF) [6,13]. All these M2 macrophage subtypes are involved in the regulation of both inflammation and tissue repair, and are differentiated by their distinct stimuli and mediators.

Regulatory macrophages (Mregs) are distinct populations of macrophages, with regulatory functions that can dampen inflammatory immune responses and suppress those of T cells [14,15]. Anti-inflammatory Mregs have therapeutic value for several diseases, such as inflammatory bowel disease (IBD; [16]), colitis [17], Crohn’s disease [18], leishmaniasis [19], and organ transplantation [15,20]. Human Mregs have been derived from CD14^+^ peripheral blood monocytes stimulated with M-CSF and then incubated with IFN-γ [21]. The induction of Mregs requires contact with a plastic surface and exposure to serum factors [20,21,22].

The most physiologically active form of vitamin D3 is 1,25-dihydroxy-vitamin D3 (vitD3), which functions in the regulation of the immune system, the nervous system, and calcium balance [23], and mediates its biological effects by binding to vitamin D receptors (VDRs) [24]. This vitamin exerts anti-inflammatory effects on monocytes/macrophages [25,26] and can induce the M2 polarization of macrophages [27,28] and microglia [29]. 

The arginine-glycine-aspartate (RGD) motif is an important structural component of various proteins that are recognized by the integrins which then mediate cell–cell interaction and adhesion to the extracellular matrix [30]. Binding to the RGD motif induces a conformational change in integrin and triggers intracellular signaling, which is involved in the regulation of macrophage adhesion and activation [31]. Because of their ability to regulate macrophage functions, both a cell adhesion protein containing the RGD motif [32] and integrin β3 [33] are responsible for the M2 polarization of macrophages. 

Herein, we have established a new protocol to induce macrophage differentiation in Mregs using vitD3 and the RGD motif without cytokines (which are typically applied to generate Mregs), and then characterized both their phenotype and their functional features.

## 2. Materials and Methods

### 2.1. Reagents and Antibodies

The following reagents were obtained from the respective suppliers: phorbol-12-myristate 13-acetate (PMA; Cayman Chemical (Ann Arbor, MI, USA)); 1,25-dihydroxy vitamin D3 (Toronto Research Chemicals, Inc., Toronto, ON, Canada); Roswell Park Memorial Institute (RPMI)-1640, Dulbecco modified Eagle medium (DMEM), Dulbecco phosphate-buffered saline (DPBS), and fetal bovine serum (FBS) (Welgene, Inc., Daegu, Republic of Korea); Interferon-gamma (IFN-γ) and anti-MER-TK antibodies (R&D Systems, Minneapolis, MN, USA); Arginylglycylaspartic acid (RGD), phytohemagglutinin P (PHA), and LPS from Escherichia coli K12 (Sigma-Aldrich, St. Louis, MO, USA); carboxyfluorescein succinimidyl ester (CFSE) CellTrace™ Cell Proliferation Kit, and antibiotic-antimycotic (Invitrogen; Thermo Fisher Scientific Inc., Waltham, MA, USA); ethylenediaminetetraacetic acid (EDTA; Amresco, Inc., Solon, OH, USA); Galα1-3Galβ1-4GlcNAcβ-PAA trisaccharide (α-Gal; GlycoTech, Gaithersburg, MD, USA); anti-CD14, anti-CD274, and anti-CD80 (eBioscience Inc. (San Diego, CA, USA); anti-CD16 (Biolegend Inc., San Diego, CA, USA); anti-DHRS9 (Abnova, Taiwan, China); anti-CD11b (BD Biosciences Inc., Franklin Lakes, NJ, USA); anti-CD209 (Serotec Bio-Rad Inc., CA, USA); and FITC-conjugated, in addition to other antibodies (Santa Cruz Biotechnology Inc., Dallas, TX, USA).

### 2.2. Cell Culture

The human monocytic cell line, THP-1 (Korean Cell Line Bank, Seoul, Republic of Korea) was grown in RPMI-1640 medium supplemented with 5% heat-inactivated fetal bovine serum (FBS) and 1% antibiotic-antimycotic. The established porcine vascular endothelial cell line MPN-3 [34] was grown in DMEM supplemented with 10% heat-inactivated fetal bovine serum (FBS) and 1% antibiotic-antimycotic solution. Jurkat T cells (E6.1) were grown in RPMI-1640 medium supplemented with 10% heat-inactivated FBS and 1% antibiotic-antimycotic. T cells were stained with 4 μM CFSE (Invitrogen) in PBS at room temperature for 10 min, washed three times with culture medium containing 10% FBS, then stimulated with 2 μg/mL PHA. The cells were maintained at 37 °C in a 5% CO2 humidified incubator.

### 2.3. Generation of Mregs 

Figure 1 shows the protocol used to generate various human macrophage subtypes. Briefly, we incubated THP-1 cells with 10 ng/mL PMA for 48 h, then stimulated them with 100 ng/mL LPS and 100 ng/mL IFN-γ to generate M1 macrophages. The PMA-induced macrophages were incubated with 1 µg/mL RGD for 24 h, followed by 1 µM vitD3 for 24 h to generate Mregs. Cells were harvested using 5 mM EDTA on ice to avoid damage by detachment, as described [35] and characterized.

### 2.4. Coculture Experiments

We assessed mitogen-induced T cell proliferation in 1 × 10^6^ Jurkat T cells incubated with 1 mL 2 μM CFSE for 10 min at 37 °C. The reaction was stopped by adding a medium containing 10% FBS, then the cells were washed with the same medium three times. Labeled cells were maintained overnight before coculture with Mregs at a ratio of 1:1. T cells and cocultured T cells + Mregs were incubated with a final concentration of 2 μg/mL PHA for 5 days, harvested, then analyzed using flow cytometry. 

We cocultured Mregs with M1 macrophages for 3 days and then assessed their inflammatory responses to xenogeneic MPN-3 cells. Cells were harvested at 3, 6, and 12 h after coculture, and the mRNA expression of inflammatory, anti-inflammatory, and coagulation-related mediators were analyzed using quantitative real-time polymerase chain reaction (qRT-PCR).

### 2.5. Preparation of RNA and qRT-qPCR

Total RNA extracted using easy-BLUETM Total RNA Extraction Kits (iNtRON Biotechnology, Inc., Seongnam-Si, Republic of Korea) as described by the manufacturer was quantified using a MaestroNano Microvolume Spectrophotometer (MaestroGen Inc., Las Vegas, NV, USA). Thereafter, cDNA was synthesized using 2 μg of total RNA in Hyperscript RT master mix (GeneAll Biotechnology, Seoul, Republic of Korea) and an Invitrogen Oligo (dT) primer (Thermo Fisher Scientific Inc.). The cDNA was amplified with qRT-PCR, using a LineGene 9600 Plus Fluorescent Quantitative Detection System (Hangzhou Bloer Technology Co., Ltd. [BIOER], Hangzhou, China) in addition to an EzAMPTM FAST One-Step RT-qPCR 2x Master Mix (SYBR; Elpis-Biotech Inc., Daejeon, Republic of Korea); the following (5′ → 3′) primer sets are shown in Table 1. 

Results were normalized, using human GAPDH as the endogenous control. The relative abundance of target mRNA in each sample was calculated from the C∆t values of the target and the GAPDH housekeeping gene, using the 2^−△△^cycle threshold (Ct) method.

### 2.6. Flow Cytometry

The cells were incubated with primary antibodies for 30 min at 4 °C, followed by either PE-conjugated or FITC-conjugated secondary antibodies at 4 °C for 30 min. After three washes with DPBS, the cells were resuspended in 0.4 mL PBS, and the cell surface protein expression was analyzed using Cytomics FC500 MLP and CXP software (Beckman Coulter Inc., Fullerton, CA, USA). 

The cells were fixed in 4% formaldehyde in DPBS, permeabilized with 0.1% Triton X-100 at room temperature for 10 min, then stained with primary anti-iNOS, anti-DHRS9, and secondary antibodies, so as to determine the expression of the intracellular protein (as described above) for the cell surface protein.

### 2.7. Statistical Analysis

All experiments were repeated at least three times. Significant differences between groups were assessed using one-way analysis of variance (ANOVA), followed by a post hoc test (Tukey’s Honestly Significant Difference) using SPSS 12.0 (SPSS Inc., Chicago, IL, USA). All data have been expressed as means ± standard deviation (SD). Differences were considered to be statistically significant at *p* < 0.05.

## 3. Results

### 3.1. Morphology and Expression of Anti-Inflammatory Cytokines in THP-1 Cells Incubated with PMA, RGD, and vitD3

One of the most important changes in Mregs differentiation is cell elongation [36]. Therefore, we examined the morphology of THP-1 cells that had been incubated with various combinations of vitD3 and RGD (Figure 1). Cells were the most elongated when incubated with 1 μg/mL RGD or 1 μM vitD3, among the assessed concentrations (Figure 2A,B). Thus, we applied this combination in subsequent experiments. The cells seemed more efficiently elongated after being incubated with PMA + RGD + vitD3 than with either PMA + RGD or PMA + vitD3 (Figure 2C). We evaluated the expression of anti-inflammatory IL-10 and pro-inflammatory IL-12 using qRT-PCR, so as to determine whether the cells generated using PMA + RGD + vitD3 had anti-inflammatory properties. Figure 3 shows the enhanced and diminished mRNA expression of IL-10 and IL-12, respectively, in these cells, indicating an anti-inflammatory phenotype.

### 3.2. Cells Incubated with PMA, RGD, and vitD3 Express Mregs Markers

Various cell surface markers are induced using M-CSF and IFN-γ in Mregs [15]. Therefore, we examined the expression of several surface markers in macrophages incubated with PMA + RGD + vitD3 using flow cytometry to determine whether these cells express the same factors as those previously established. Generic markers of human macrophages, CD11b, CD16, and CD33, were expressed, whereas CD14 expression was low (Figure 4). Typical markers of the tissue-resident macrophages CD163, CD206, CD209, and MER-TK were significantly enhanced in cells that had been incubated with PMA + RGD + vitD3 (Figure 4). The expression of most surface markers, including costimulatory CD80 and CD86 and inhibitory CD274 and TREM-2, were similar between macrophages that had been incubated with PMA + RGD + vitD3 and established Mregs [15]. Significantly less iNOS was expressed in macrophages that had been incubated with PMA + RGD + vitD3 than in resting macrophages (Figure 4). Human Mregs selectively express the stable marker dehydrogenase/reductase 9 (DHRS9), which is involved in the initial step of synthesizing the important immune regulator retinoic acid from retinol or β-carotene, ingested as a vital nutrient [22]. We found more abundant DHRS9 expression in macrophages incubated with PMA + RGD + vitD3 than in the other cells examined (Figure 5A,B). The elongated cells and surface marker expression indicated that the macrophages-induced PMA + RGD + vitD3 have the morphological and phenotypic characteristics of Mregs.

### 3.3. Mregs Stably Express Mregs Markers Even When Exposed to Inflammatory Stimulants

The phenotypic and functional stability of Mregs are crucial factors for their clinical application, and their instability under inflammatory conditions is a problem that should be addressed [12]. Since Mregs polarization can depend on IFN-γ [22], we examined DHRS9 expression in Mregs that had been incubated with IFN-γ. We found that IFN-γ did not affect the enhanced expression of DHRS9 in Mregs, indicating that the stability of Mregs remained unchanged with or without IFN-γ (Figure 5C,D). We further investigated the phenotypic stability of several Mregs markers in a pro-inflammatory environment, using flow cytometry. We incubated the Mregs with LPS, a major component of bacterial cell walls that stimulates macrophages to generate inflammatory responses. The enhanced expression of the Mregs markers CD163, CD206, CD209, CD274, and DHRS9 (Figure 6A,B), and the decreased expression of the inflammatory markers, CD80, iNOS, TLR2, and TLR4 (Figure 6C,D), were not affected in Mregs that had been incubated with LPS. We then examined the expression of several markers in Mregs that had been exposed to the xenogeneic antigen α-Gal, which is a sugar molecule that is expressed in porcine (but not human) cells, and thus directly activates human macrophages [37,38]. The expression of markers in Mregs incubated with or without α-Gal did not differ (Figure 7A). The expression of the inflammatory markers CD80 and iNOS were significantly enhanced by α-Gal in THP-1 cells compared with control cells incubated without α-Gal, whereas that of Mregs did not differ, regardless of α-Gal (Figure 7B).

### 3.4. Expression of Pro-Inflammatory and Immunosuppressive Mediators Is Respectively Damped and Enhanced in Mregs Cocultured with Xenogeneic Cells

We assessed the mRNA expression of pro-inflammatory and immunosuppressive mediators in THP-1 cells, the M1 macrophages derived from THP-1 cells (in accordance with the procedure described in Figure 1), and the Mregs cocultured with the porcine aortic vascular endothelial cell line MPN3 [34]. Expression of the pro-inflammatory mediators IL-1β, IL-6, IL-12, TNF-α, MCP-1, and iNOS was significantly enhanced, whereas that of anti-inflammatory IL-10 was significantly decreased in cocultured M1 macrophages and MPN3, compared with cocultured THP-1 and MPN3 cells. In contrast, the expression of pro-inflammatory mediators was significantly suppressed, whereas that of the immunosuppressive mediators IL-10 and IDO was obviously enhanced in cocultured Mregs and MPN3 cells (Figure 8A). The coagulation-associated mediators fgl2, PAR-1, and TF play important roles in inflammation and coagulation during xenograft rejection [39]. We examined the mRNA expression of these factors in THP-1 cells, M1 macrophages, and Mregs cocultured with MPN3 cells. The expression of human fgl2, PAR-1, and TF was significantly more increased in M1 and MPN3 than it was in THP-1 and MPN3 cocultures, and significantly lower in cocultured Mregs and MPN3 than it was in the other two groups (Figure 8B). We cocultured Mregs with M1 and MPN3 cells at different ratios and assessed the mRNA expression of pro-inflammatory and coagulation-associated mediators, so as to determine whether Mregs would suppress immune responses. The expression of pro-inflammatory mediators was significantly increased in M1 and MPN3, compared with THP-1 and MPN3 cocultures (Figure 9A). However, the enhanced expression was completely inhibited at a 1:1 ratio of Mregs:M1. The expression of inflammatory mediators gradually increased as the Mregs:M1 ratio decreased, reaching the levels of M1 only at a Mregs/M1 ratio of 1:4, except for MCP-1, the expression of which was completely suppressed at this ratio (Figure 9A). Similarly, the expression of coagulation-associated mediators was significantly increased in M1 and MPN3 compared with THP-1 and MPN3 cocultures (Figure 9B). However, this was completely inhibited at a 1:1 ratio of Mregs to M1. The expression of coagulation-associated mediators gradually increased as the Mregs:M1 ratio decreased, and reached the highest levels only when the Mregs/M1 ratio was 1:4. The expression for TF, however, remained significantly suppressed at this ratio (Figure 9B). These results indicated that our Mregs suppress the expression of inflammatory and coagulation-associated mediators of M1 macrophages in response to xenogeneic α-Gal and bacterial LPS.

### 3.5. Regulatory Macrophages Immunosuppress Mitogen-Induced T Cell Proliferation

The suppression of mitogen-induced T cell proliferation is a distinct feature of Mregs [21]. Therefore, we investigated whether Mregs could suppress the mitogen-stimulated Jurkat T cell line. We cocultured Mregs with CFSE-labeled Jurkat T cells and incubated them with the T cell mitogen known as phytohemagglutinin (PHA). Figure 10 shows that T cells stimulated with PHA rapidly proliferated (gray shadow), whereas those stimulated with a 1:1 ratio of Mregs to T cells did not proliferate (red line). These results indicated that Mregs suppressed mitogen-induced T cell proliferation.

## 4. Discussion

We induced Mregs from THP-1 monocytes that had been incubated with PMA, RGD and vitD3. We did not use M-CSF and IFN-γ, which are commonly used to generate human Mregs [11]. Our Mregs had an elongated morphology, which was in line with other findings [15,40], and were phenotypically characterized as CD11b^hi^, CD14^lo^, CD16^+^, CD33^+^, CD80^lo^, CD86^+^, iNOS^-^, CD163^hi^, CD206^+^, CD209^+^, CD274^+/hi^, TREM-2^+^, MER-TK^hi^, and DHRS9^+/hi^. Most of the markers evaluated herein were expressed in the Mregs generated by others, including CD14, CD80, CD86 [15,21,41], CD163, CD206, CD274 [15,42], CD209, MER-TK [15], and DHRS9 [22]. Among these, DHRS9 was considered a specific and stable marker of Mregs because it was not expressed by other known human monocyte-derived tolerogenic or suppressive cells used for cell-based immunotherapies [22], such as immature monocyte-derived dendritic cells (DCs), tolerogenic DCs [43], rapamycin-treated DCs [44], IL-10 conditioned DCs [45], or prostaglandin E2-induced myeloid-derived suppressor cells [46]. Furthermore, DHRS9 expression was stable even when Mregs were exposed to LPS or IFN-γ [22]. Herein, DHRS9 expression was stable in Mregs that had been exposed to IFN-γ, LPS, and xenogeneic α-Gal. Our Mregs had anti-inflammatory phenotypes, with upregulated IL-10 and downregulated IL-12, iNOS, CD80, TLR2, and TLR4 expression. In addition, the expression of anti-inflammatory mediators was enhanced, while those of pro-inflammatory and coagulation-related mediators were damped in response to xenogeneic antigens or cells. Furthermore, Mregs inhibited mitogen-induced T-cell proliferation, which was another cardinal feature of Mregs [21]. Taken together, the present findings showed that incubating THP-1 monocytes with PMA + RGD + vitD3 induced functional Mregs.

Regulatory macrophages can be induced with combinations of M-CSF and IFN-γ, LPS and immune complex, LPS and prostaglandin E2, or anti-TNF-α monoclonal antibody and others [11]. Among them, M-CSF + IFN-γ was the most prevalent inducer, especially of human Mregs. The induction of Mregs normally requires two consecutive signals; primary signals generated with M-CSF that induce monocyte differentiation into macrophages [47], and secondary signals stimulated with IFN-γ that induce Mregs to harness anti-inflammatory properties [48]. Because PMA can induce the monocyte’s differentiation into macrophages [49], it might have acted like M-CSF in the process of Mregs induction in the present study. Because vitD3 exerts anti-inflammatory effects on macrophages [25,26] and induces M2 polarization of macrophages through the VDR signaling mechanism [27,28], we speculate that it acts as a stimulant for secondary signals to coordinate an anti-inflammatory environment. We also used RGD to generate Mregs in addition to these two reagents. The RGD sequence represents the principal integrin-binding motif found in extracellular matrix (ECM) proteins, such as fibronectin [50]. Macrophages, compared to other immune cells, rely more heavily on a continuous interaction with the ECM, particularly through integrin-fibronectin binding (or its RGD motif), in order to develop specific phenotypes and functions. For instance, this interaction leads to macrophage elongation and subsequent anti-inflammatory functions [51]. During inflammation, the ECM undergoes extensive remodeling, providing crucial signals for macrophage migration, activation, and the resolution of inflammation to promote wound healing [52].

Cell adhesion proteins containing the RGD motif can promote M2 polarization by activating macrophage integrins through their RGD motif, suggesting that RGD-integrin-induced cellular signaling regulated the M2 polarization [32]. Similarly, in an M2-favorable environment, the RGD motif facilitated the M2 polarization of the macrophages [53,54]. In our recent study, we demonstrated that THP-1 cells cultured on RGD- and polydopamine-coated micropatterns were able to induce their differentiation into Mregs-like cells, even in the absence of cytokines [40]. Additionally, integrin (which is a counterpart to RGD) was also involved in M2 polarization. Increased expression of the integrins α4 [55] and β3 [33] was associated with M2-like characteristics. Furthermore, M2 polarization was dependent on integrin β3 through the transcription factor peroxisome proliferator-activated receptorγ (PPARγ), which interacted with VDRs [33]. Taken together, these results suggest that the RGD motif facilitated the Mregs polarization of PMA-induced macrophages in a vitD3-induced anti-inflammatory environment.

Regulatory macrophages exert immunosuppressive effects in human kidney transplantation. Hutchinson et al. found that administering donor Mregs to kidney recipients dampened the allogeneic rejection response, thus minimizing the need for conventional immunosuppressive therapy [41,42]. One mechanism of allograft acceptance in recipients is Mregs-induced Treg generation [15,21]. Macrophages mainly reside in the medullary cords of lymph nodes that are adjacent to the paracortical area where most T cells reside. Therefore, they probably interact intimately and influence each other in terms of functional activity. Macrophages that induce regulatory T cells are assumed to be Mregs [56]). In turn, interactions between Tregs and macrophages can result in polarization of the latter into regulatory phenotypes [57]. Thus, the ability of Mregs to suppress T-cell proliferation might have been due to their effects on Treg expansion and immunosuppressive cytokine production [14,15].

To some extent, our research had several limitations. One of the key limitations was the omission of an examination into the Mregs’ function in generating regulatory T cells. To address this limitation, in further studies, we will need to validate the Mregs’ function using primary cells, such as PBMCs or BMDMs and animal models, before considering their potential use in clinical trials. Additionally, it is important to note that our long-term objective is to develop an effective and affordable therapy for patients. To achieve this, we need to refine our protocols in order to generate an adequate quantity of cells, as well as establish a sustainable production process over an extended period of time. This optimization is crucial if we are to ensure that the therapy can be scaled up and readily available for clinical use.

## 5. Conclusions

We showed that sequential incubations with PMA, RGD and vitD3 could generate human Mregs from THP-1 monocytes. Our Mregs expressed Mregs markers and had both anti-inflammatory and immunosuppressive properties. Our method was simple and cost-effective because it did not require cytokines. However, our study had some limitations. We did not evaluate the capacity of Mregs to induce Treg expansion, and used an established cell line instead of primary cells to generate Mregs. Thus, further studies are needed to elucidate the epigenetic and metabolic reprogramming that occurs during macrophage polarization, so as to determine the mechanisms underlying Mregs induction. In addition, Mregs should be established using primary human monocytes, in order to evaluate their possible application in various inflammatory states, including endotoxemia and organ transplantation.

## Figures and Tables

**Figure 1 biomedicines-11-01740-f001:**
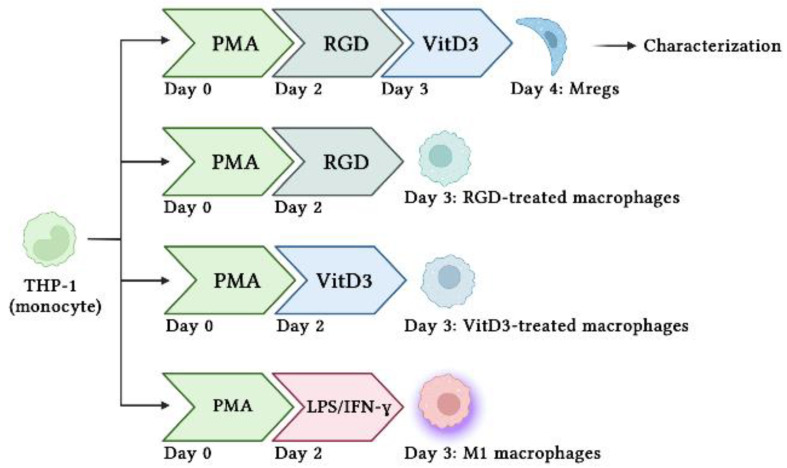
Schema of procedures to generate different types of macrophages using human THP-1 monocytes.

**Figure 2 biomedicines-11-01740-f002:**
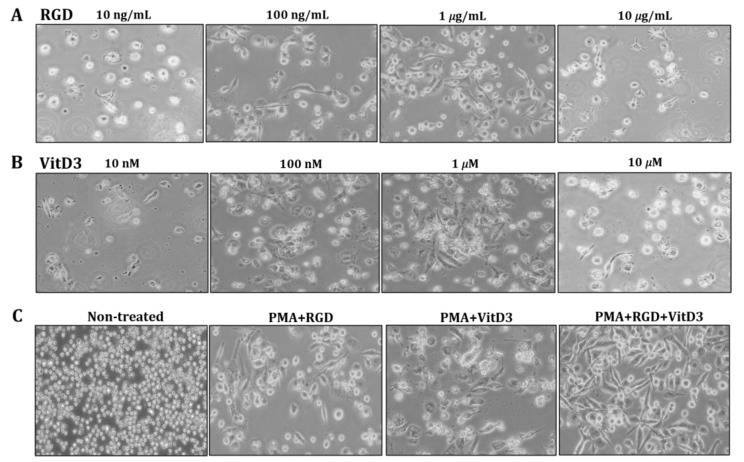
Incubation with PMA, RGD, and vitD3 induces THP-1 cell elongation. We incubated THP-1 cells with PMA (10 ng/mL) for 2 days, followed by various concentrations of RGD (**A**) or vitD3 (**B**) for 1 day. Cells were incubated with PMA (10 ng/mL) for 2 days, followed by RGD (1 μg/mL), then 1 μM vitD3 for 1 day (PMA + RGD + VitD3 group; (**C**) Cell morphology was visualized using an inverted microscope at 100× magnification.

**Figure 3 biomedicines-11-01740-f003:**
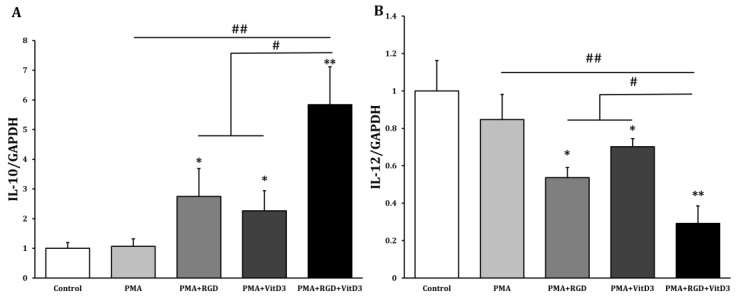
Incubation with PMA, RGD, and vitD3 upregulated IL-10 and downregulated IL-12 in THP-1 cells. We incubated THP-1 cells with 10 ng/mL PMA for 2 days, washed them with DPBS, then maintained them in RPMI-1640 media for 2 days (PMA group). Cells were incubated with PMA (10 ng/mL) for 2 days, followed by RGD (1 μg/mL; PMA + RGD) or vitD3 (1 μM; PMA + VitD3) for 1 day. Cells were washed with DPBS, then maintained in RPMI-1640 media for 1 day. We incubated cells with PMA (10 ng/mL) for 2 days, followed (sequentially) by RGD (1 μg/mL) for 1 day and vitD3 (1 μM) for 1 day (PMA + RGD + VitD3). Cells were harvested, and the mRNA expressions of anti-inflammatory IL-10 (**A**) and pro-inflammatory IL-12 (**B**) were quantified using qRT-PCR. * *p* < 0.05, ** *p* < 0.005 vs. controls incubated with DMSO; # *p* < 0.05, ## *p* < 0.005.

**Figure 4 biomedicines-11-01740-f004:**
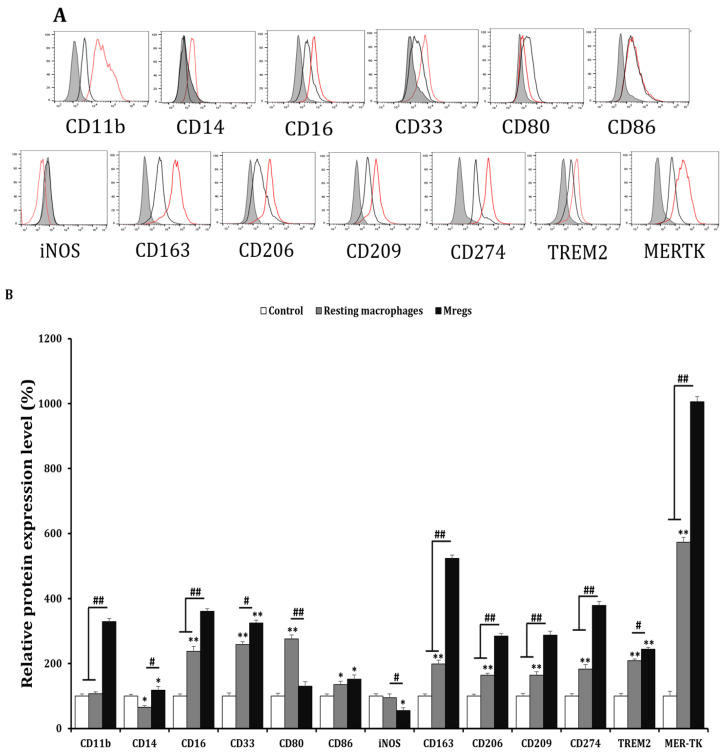
Phenotypic characterization of PMA/RGD/vitD3-induced Mregs-like cells. We generated resting macrophages by incubating THP-1 cells with PMA (10 ng/mL) for 2 days, washing them with DPBS, then maintaining them in RPMI-1640 media for 2 days (resting macrophages group). Cells were incubated with PMA (10 ng/mL) for 2 days, followed by RGD (1 μg/mL) for 1 day, then with vitD3 (1 μM) for 1 day to generate Mregs. Cells were harvested, and the expression of the cell surface and intracellular markers were assessed using flow cytometry. (**A**) Black lines, red lines, and gray shadows on representative flow cytometric histograms indicate resting macrophages, Mregs, and controls, respectively. (**B**) Bar graphs show the mean values of relative protein expression ± SD. * *p* < 0.05, ** *p* < 0.005 vs. DMSO controls. # *p* < 0.05, ## *p* < 0.005.

**Figure 5 biomedicines-11-01740-f005:**
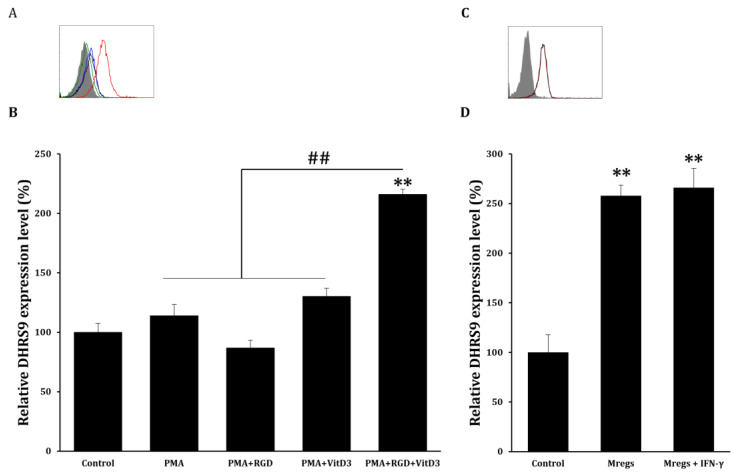
DHRS9 expression in Mregs. (**A**,**B**) cells were incubated with PMA (10 ng/mL) for 2 days, washed with DPBS, then maintained in RPMI-1640 media for 2 days (PMA group). Cells were incubated with PMA (10 ng/mL) for 2 days, followed by RGD (1 μg/mL; PMA+ RGD group) or vitD3 (1 μM; PMA + VitD3 group) for 1 day, washed with DPBS, then maintained in RPMI-1640 media for 1 day. Cells were incubated with PMA (10 ng/mL) for 2 days, followed by RGD (1 μg/mL) for 1 day, then 1 μM vitD3 for 1 day (PMA + RGD + VitD3). Cells were harvested and DHRS9 expression was quantified using flow cytometry. (**A**) Blue, green, black, and red lines in representative flow cytometric histograms indicate PMA, PMA + RGD, PMA + VitD3, and PMA + RGD + VitD3 (Mregs), respectively. Gray shadows indicate the control. (**B**) Bar graphs show the mean values of relative DHRS9 expression ± SD. (**C**,**D**) Mregs were incubated with IFN-γ (25 ng/mL) for 24 h, and DHRS9 expression was measured using flow cytometry. (**C**) Red and black lines in representative flow cytometric histograms indicate Mregs and Mregs + IFN-γ, respectively. Gray shadow indicates the control. (**D**) Bar graphs show the mean values of relative DHRS9 expression ± SD. ** *p* < 0.005 vs. DMSO controls; ## *p* < 0.005.

**Figure 6 biomedicines-11-01740-f006:**
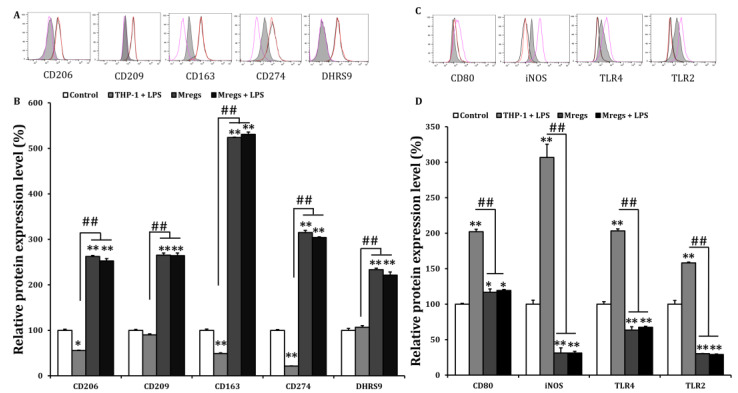
Marker expression of Mregs did not change upon exposure to LPS. (**A**–**D**) We generated Mregs as described in Figure 5. We stimulated THP-1 cells or Mregs with LPS (100 ng/mL) for 24 h, then quantified expression of cell surface markers using flow cytometry. For (**A**,**C**), the red and black lines are representative of the flow cytometric histograms for Mregs and Mregs + LPS, respectively. Pink lines and gray shadow indicate THP-1 + LPS and controls, respectively. For (**B**,**D**), Bar graphs show the mean values of relative protein expression ± SD. * *p* < 0.05, ** *p* < 0.005 vs. controls incubated with DMSO; ## *p* < 0.005.

**Figure 7 biomedicines-11-01740-f007:**
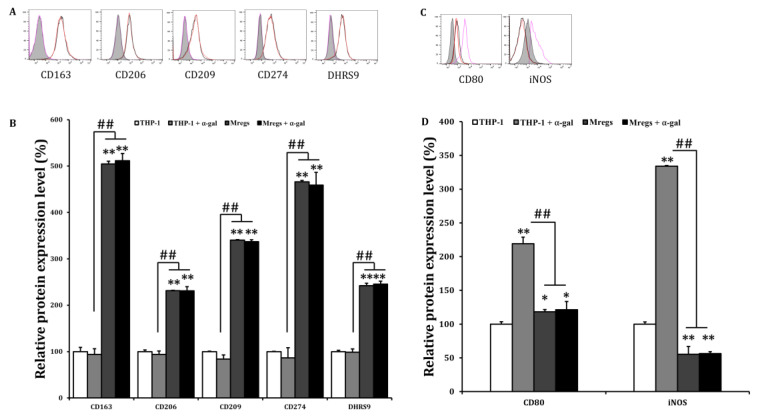
Marker expression of Mregs did not change upon exposure to xenogeneic antigen. (**A**–**D**) We generated Mregs as described in Figure 5. We stimulated THP-1 cells or Mregs with α-gal (100 ng/mL) for 24 h, then quantified the expression of cell surface markers using flow cytometry. (**A**,**C**) are the representative flow cytometric histograms, for which red and black lines indicate Mregs and Mregs + α-gal, respectively, while pink lines and gray shadow indicate THP-1 + α-gal and controls, respectively. (**B**,**D**) are the mean values of relative protein expression ± SD. * *p* < 0.05, ** *p* < 0.005 vs. controls incubated with DMSO; ## *p* < 0.005.

**Figure 8 biomedicines-11-01740-f008:**
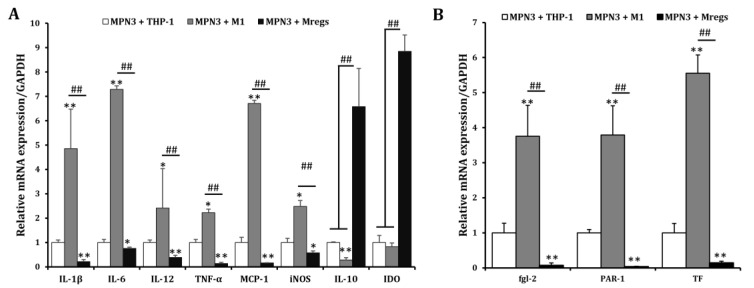
Regulatory macrophages expressed fewer inflammatory and coagulation-related genes and more immunosuppressive genes in response to pig endothelial cell line MPN3. We generated M1 macrophages by incubating THP-1 cells with LPS (100 ng/mL) and IFN-γ (20 ng/mL) for 24 h, and generated Mregs as described in Figure 5. Thereafter, MPN-3 cells were cocultured with THP-1, M1, or Mregs, for 2 h (fgl-2, and TF), 6 h (IDO, IL-1β, IL-6, IL-10, IL-12, iNOS, and TNF-α), or 12 h (PAR-1 and MCP-1), respectively. Cells were harvested, and the mRNA expression of cytokines and chemokines (**A**), as well as coagulation-related genes (**B**), were evaluated using qRT-PCR. * *p* < 0.05, ** *p* < 0.005 vs. MPN3 + THP-1; ## *p* < 0.005.

**Figure 9 biomedicines-11-01740-f009:**
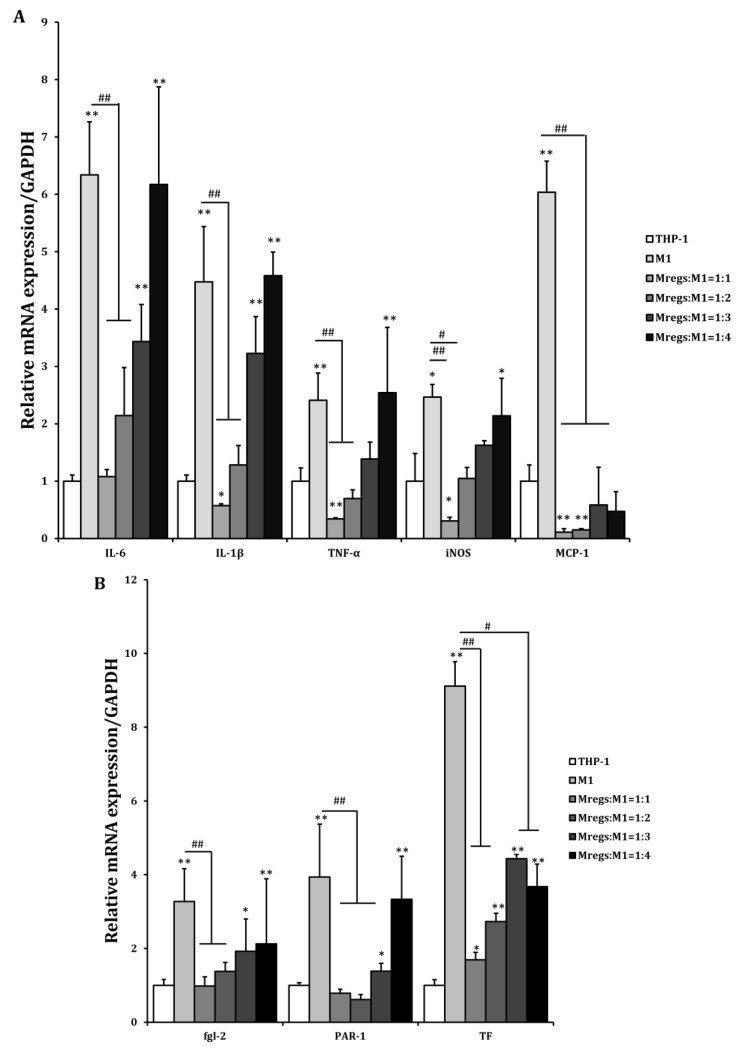
Regulatory macrophages suppressed mRNA expression of pro-inflammatory and coagulation-related genes in M1 and MPN3 cocultures, according to the ratios of Mregs and M1 cells. We generated M1 macrophages by incubating THP-1 cells with LPS (100 ng/mL) and IFN-γ (20 ng/mL) for 24 h, in order to generate Mregs as described in Figure 5. Thereafter, Mregs were cocultured them at indicated ratios for 1 day. Thereafter, THP-1, M1, or Mregs/M1 cocultures were incubated with MPN-3 cells for 2 h (fgl-2, and TF), 6 h (IDO, IL-1β, IL-6, IL-10, IL-12, iNOS, and TNF-α), or 12 h (PAR-1 and MCP-1). Cells were harvested and mRNA expression of cytokines and chemokines (**A**) and coagulation-related genes (**B**) were evaluated using qRT-PCR. * *p* < 0.05, ** *p* < 0.005 vs. MPN3 + THP-1; # *p* < 0.05, ## *p* < 0.005.

**Figure 10 biomedicines-11-01740-f010:**
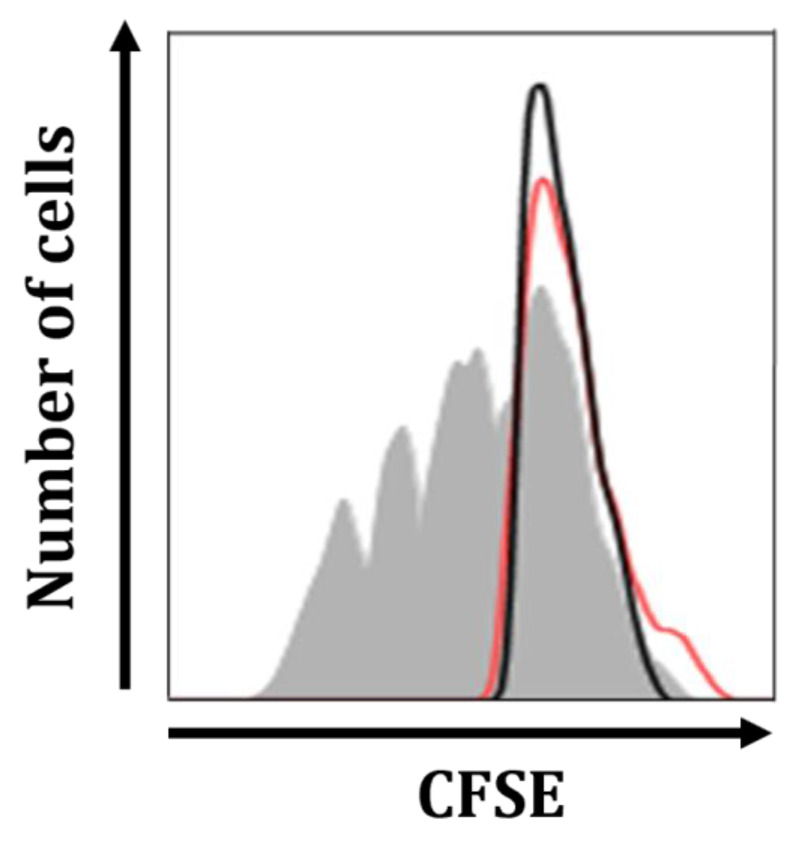
Mregs suppressed mitogen-induced proliferation of Jurkat T cells. We generated Mregs as described in Figure 5, then cocultured them with CFSE-labeled Jurkat T cells at a 1:1 ratio. Cocultured and T cell groups (5 days) were stimulated with PHA (2 µg/mL) for 5 days, then harvested and quantified using flow cytometry. The gray shadow, black line, and red line in representative histograms show T cells on day 5, day 0, and cocultured Mregs and T cells on day 5, respectively.

**Table 1 biomedicines-11-01740-t001:** Primer sets used in this study.

fgl-2	F: AGCTGATGACAGCAGAGTTAGAG
	R: AGTGATCATACAAGGCATAGAGC
iNOS	F: ACAGCACATTCAGATCCCCA
	R: AACACGTTCTTGGCATGCAT
IL-1β	F: GGGATAACGAGGCTTATGTGC
	R: AGGTGGAGAGCTTTCAGTTCA
IL-6	F: GACCCAACCACAAATGCCAG
	R: GAGTTGTCATGTCCTGCAGC
IL-10	F: TCTCCGAGATGCCTTCAGCAGA
	R: TCAGACAAGGCTTGGCAACCCA
IL-12	F: ACGAGTGCTCCTGGCAGTAT
	R: AGGATTCCACCCAGAGTGTG
IDO	F: TGCAAGAACGGGACACTTTG
	R: CCCTTCATACACCAGACCGT
MCP-1	F: CCCAAGAATCTGCAGCTAAC
	R: GGTAGAACTGTGGTTCAAGAGG
TF	F: GGGCTGACTTCAATCCATGT
	R: GAAGGTGCCCAGAATACCAA
TNF-α	F: TGAGCACTGAAAGCATGATCC
	R: GGAGAAGAGGCTGAGGAACA
PAR-1	F: CATCTGTGTACACCGG
	R: TGCCAATCACTGCC
GAPDH	F: ACAGCCTCAAGATCATCAGCAAT
	R: AGGAAATGAGCTTGACAAAGTGG

The abbreviations listed correspond to various genes or proteins, such as Fibrinogen-like protein 2 (fgl-2), inducible nitric oxide synthase (iNOS), indoleamine 2,3-dioxygenase (IDO), monocyte chemoattractant protein-1 (MCP-1), tissue Factor (TF), tissue necrosis factor alpha (TNF-α), protease-activated receptor-1 (PAR-1), and glycer-aldehyde-3-phosphate dehydrogenase (GAPDH).

## Data Availability

The data presented in this study are available on request from the corresponding author.

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
