# Peer review of "Human Regulatory Macrophages Derived from THP-1 Cells Using Arginylglycylaspartic Acid and Vitamin D3"

_biomedicines, 2023, doi:10.3390/biomedicines11061740_

Round 1

Reviewer 1 Report

Pham et al. present in this paper a novel differentiation pathway to generate regulatory macrophages (Mreg). Polarization of macrophages is an important field to better understand immune-regulation and to develop novel therapeutic tools. The authors use vitamin D3 and RGD-motif to induce potent inhibitory and anti-inflammatory Mregs. This is of course an interesting finding. Yet, all experiments were performed with cell lines (THP1 and Jurkat). Although, both cell types are widely used in immunological studies, it is nowadays essential to confirm findings obtained with tumor-derived cell lines in primary human cells, i.e. monocytes and T cells. 

Author Response

Thank you for your kind and considerate comments, which have greatly helped improve the quality of our current version. We have carefully studied your comments, and we would like to explain that this is our initial step to explore a new combination to generate Mregs. In future experiments, we plan to verify these findings based on primary cells and then in an in vivo model. We hope this answer meets your requirements.

Reviewer 2 Report

Pham et al. describes a novel method of generating regulatory macrophages without using typical cytokines such as IFNγ. Generating Mregs from THP-1 cells using small molecules allows for a reproducible, inexpensive and elegant way for Mreg expansion for in vitro experimentation. The work has also validated the characteristics of the macrophages generated by their novel protocol, finding several hallmark characteristics that resemble that of earlier reports on Mreg. The methods are well described, and the characterization has been done to a reasonable extent to allow for the findings and claims in this work. I find that the work is of high enough quality that warrants publication in Biomedicines.

One minor comment is to include in the discussion section, the importance of macrophage shape on their function. Just like macrophages are known to adopt several different morphologies when they are stimulated with contrasting polarizing stimuli, the shape they adopt can also influence their phenotype. One work (already cited in the paper, ref 36 - McWhorter et al.) has described that macrophages when elongated by cell culture on micropatterned fibronectin lines show anti-inflammatory M2 characteristics. Similarly, another relevant work to cite in the discussion section is https://doi.org/10.1016/j.biomaterials.2021.121236 which has shown that macrophage elongation on fibronectin lines can also help them resist M1 stimuli induced inflammation. That study also describes an ECM-mediated mechanism that can affect pro-inflammatory epigenetic factors. Given that the work by Pham et al. show the importance of RGD molecules in Mreg induction, it may be possible that Mreg induction in vivo is a consequence of modulation of lineage defining epigenetic and transcriptional factors through microenvironmental (biomechanical/ECM) cues.

Author Response

Thank you for your kind and considerate comments on our manuscript. The authors have carefully studied your comments and have made an effort to discuss the importance of macrophage morphology in regulating their function, in line with your requirements. The revised portions are highlighted in the revised manuscript for your kind consideration.

Round 2

Reviewer 1 Report

The response by the authors is really concise. I somehow understand that it is probably to much to repeat everything with primary cells but I could not find any sentence to mention this problem or weakness of the paper at least in the discussion. 

Author Response

Dear Reviewer, we appreciate your precious time in revising our manuscript and providing us insightful comments that help improve the quality of our paper. We aslo want to apologize for our mistake that we did not mention limitations of our study. In this version of our manuscript, we have carefully completed our discussion, which we hope to meet your approval.